# Pasture Biomass Estimation Using Ultra-High-Resolution RGB UAVs Images and Deep Learning

**Milad Vahidi, Sanaz Shafian \*, Summer Thomas and Rory Maguire**

School of Plant and Environmental Sciences, Virginia Polytechnic Institute and State University, Blacksburg, VA 24061, USA; miladvahidi@vt.edu (M.V.); sumtho@vt.edu (S.T.); rmaguire@vt.edu (R.M.)
\* Correspondence: sshafian@vt.edu

**Abstract:** The continuous assessment of grassland biomass during the growth season plays a vital role in making informed, location-specific management choices. The implementation of precision agriculture techniques can facilitate and enhance these decision-making processes. Nonetheless, precision agriculture depends on the availability of prompt and precise data pertaining to plant characteristics, necessitating both high spatial and temporal resolutions. Utilizing structural and spectral attributes extracted from low-cost sensors on unmanned aerial vehicles (UAVs) presents a promising non-invasive method to evaluate plant traits, including above-ground biomass and plant height. Therefore, the main objective was to develop an artificial neural network capable of estimating pasture biomass by using UAV RGB images and the canopy height models (CHM) during the growing season over three common types of paddocks: Rest, bale grazing, and sacrifice. Subsequently, this study first explored the variation of structural and color-related features derived from statistics of CHM and RGB image values under different levels of plant growth. Then, an ANN model was trained for accurate biomass volume estimation based on a rigorous assessment employing statistical criteria and ground observations. The model demonstrated a high level of precision, yielding a coefficient of determination ($R^2$) of 0.94 and a root mean square error (RMSE) of 62 ($g/m^2$). The evaluation underscores the critical role of ultra-high-resolution photogrammetric CHMs and red, green, and blue (RGB) values in capturing meaningful variations and enhancing the model's accuracy across diverse paddock types, including bale grazing, rest, and sacrifice paddocks. Furthermore, the model's sensitivity to areas with minimal or virtually absent biomass during the plant growth period is visually demonstrated in the generated maps. Notably, it effectively discerned low-biomass regions in bale grazing paddocks and areas with reduced biomass impact in sacrifice paddocks compared to other types. These findings highlight the model's versatility in estimating biomass across a range of scenarios, making it well suited for deployment across various paddock types and environmental conditions.

**Keywords:** biomass estimation; RGB images; structural variables; spectral variables; paddock types; learning algorithms

## 1. Introduction

Estimating pasture biomass is essential in agricultural management, providing crucial insights for promoting sustainable livestock production. It plays a key role in refining grazing practices, determining optimal animal stocking rates, and preventing excessive land use that can lead to soil degradation and reduced agricultural output. Accurate biomass estimation empowers farmers to make wise decisions about when and how much livestock to graze, ultimately enhancing animal yield and profitability while preserving pasture ecosystems' health and ecological equilibrium. Furthermore, precise biomass estimation is a valuable resource for land stewards and policymakers, offering insights into the feasibility of sustainable livestock production and the effective management of natural resources [1–3].

Satellite remote sensing technologies have proven to be a promising tool in estimating pasture biomass. An exemplary study by Alckmin et al. (2008) leveraged 1 m hyperspectral remote sensing data to estimate grass biomass, demonstrating precision and the potential to monitor changes in pasture productivity over time [4]. Subsequent studies have highlighted the effectiveness of satellite-based imagery in accurately quantifying pasture biomass across various management scenarios. For instance, Dusseux et al. (2018) [5] used Sentinel-2 satellite imagery (at a 10 m spatial resolution and the six spectral bands at 20 m) to assess pasture biomass within French dairy farms, achieving an impressive accuracy rate of up to 90%. Similarly, Peng et al. (2023) [6] applied different regression models on Sentinel-2 images (at a 10 m spatial resolution and the six spectral bands at 20 m) to estimate biomass within forage fields in northern Sweden, yielding a notable accuracy level of 92%. Moreover, Milad et al. applied an integration Sentinel product (10 m Sentinel-1 and Sentinel-2 images) and different learning tools to estimate tall fescue pasture biomass over different paddock types. The developed model could estimate pasture biomass with an accuracy of 0.83 [7].

While satellite remote sensing techniques have yielded promising results in estimating pasture biomass, it is essential to acknowledge the persisting challenges associated with achieving accurate biomass estimation in specific variations of pasture systems. For instance, heterogeneous vegetation or non-vegetated areas can introduce complexity into biomass estimation, as demonstrated by previous research [8]. Furthermore, limitations in the availability and adequacy of ground-level measurements and calibration data frequently constrain biomass estimation accuracy derived from satellite imagery [9]. Moreover, satellite images are susceptible to cloud cover, impairing their availability and temporal resolution. These images can also be influenced by atmospheric conditions, such as haze or aerosols, thereby diminishing the reliability of biomass estimations [10].

To address the limitations associated with satellite-based imagery, researchers have directed their attention toward utilizing unmanned aerial vehicle (UAV) imagery for biomass estimation [11–14]. UAVs offer flexibility by capturing images from varied perspectives and diverse lighting conditions, thereby mitigating the impact of shadows and other environmental variables on biomass estimations [15]. Utilizing UAV-based imagery has demonstrated cost-effectiveness, especially for small to medium-sized farms, reducing the need for costly equipment or specialized personnel. Moreover, UAVs can swiftly cover expansive areas, facilitating rapid data acquisition and analysis—an especially crucial factor for monitoring fluctuations in pasture productivity over time.

Most of the research conducted on pasture biomass estimation using UAV-based images encompasses several distinct dimensions. Some studies focused on the nature of variables extracted from UAV images, such as structural variables [16–18], spectral variables [19,20], or their combinations [21,22]. Another group of scientists has focused on the mathematical underpinnings of modeling, such as parametric models [23,24] or nonparametric models [25,26]. The third group has focused on applying various learning tools, the design of experiments, and the meticulous planning of flight missions [27].

Research studies that exclusively concentrated on utilizing UAV-based spectral variables for pasture biomass estimation generated algorithms with limited performance, resulting in relatively modest $R^2$ values, typically falling within the range of 0.45 to 0.72 [28]. The other studies, which exclusively relied on UAV-based structural variables such as digital surface models (DSM) and digital terrain models (DTM), did not consistently achieve the desired levels of accuracy, yielding $R^2$ values within the range of 0.54 to 0.81 [28].

It is noteworthy that our comprehensive literature review revealed several studies that successfully integrated UAV-based structural and spectral variables for biomass estimation. However, a crucial gap persists, as no studies to date have integrated the spectral, structural, and categorical variables extracted from UAV time-series images using machine learning tools for estimating biomass across diverse types of pasture paddocks. The inclusion of categorical variables, such as pasture type, is paramount for refining biomass estimation models and enhancing accuracy. Recognizing the significance of factors beyond spectral and structural variables, integrating categorical variables into the model introduces a

nuanced understanding of biomass dynamics. In this study, we underscore the importance of pasture type as a key variable, aiming to elevate the accuracy of biomass estimation. By incorporating this categorical dimension alongside spectral and structural data from UAV time-series images, we anticipate a more comprehensive and precise model that captures the intricacies of biomass variation across different pasture paddocks. This novel approach holds the potential to yield insights into nuanced variations in biomass patterns, contributing to a more refined and adaptable model for sustainable pasture management. Our study seeks to fill this critical research gap and advance the integration of cutting-edge technologies and methodologies in the field of pasture biomass estimation. We also aim to advance the field by crafting an advanced deep-learning model explicitly designed for pasture biomass estimation. This cutting-edge approach can potentially enhance predictive accuracy, which was not implemented before significantly. Finally, we will apply the developed model to a series of UAV images taken over diverse paddock types, including bale grazing, sacrificed paddocks, and rest paddocks, and estimate pasture biomass. This systematic approach to experimental design, including the temporal element and diverse paddock types, marks a departure from previous studies.

## 2. Materials

### 2.1. Study Area

This study was carried out throughout the spring and summer of 2022 at the Shenandoah Valley Agricultural Research and Extension Center (SVAREC) in Virginia. The center's research and extension projects address cattle productivity, forage systems, small-scale forestry, and wood lot management. The study area, positioned at a geographical center of 37.934711°N and −79.216526°W, is nestled within an elevation span of 500 to 580 m above sea level. Geometrically, the center spans over 900 acres, with diverse topography ranging from a 50% slope to flat plains. The predominant vegetation type is pasture for cattle grazing. The geographical layout of the case study, including paddocks and the region's perimeter, is illustrated in Figure 1.

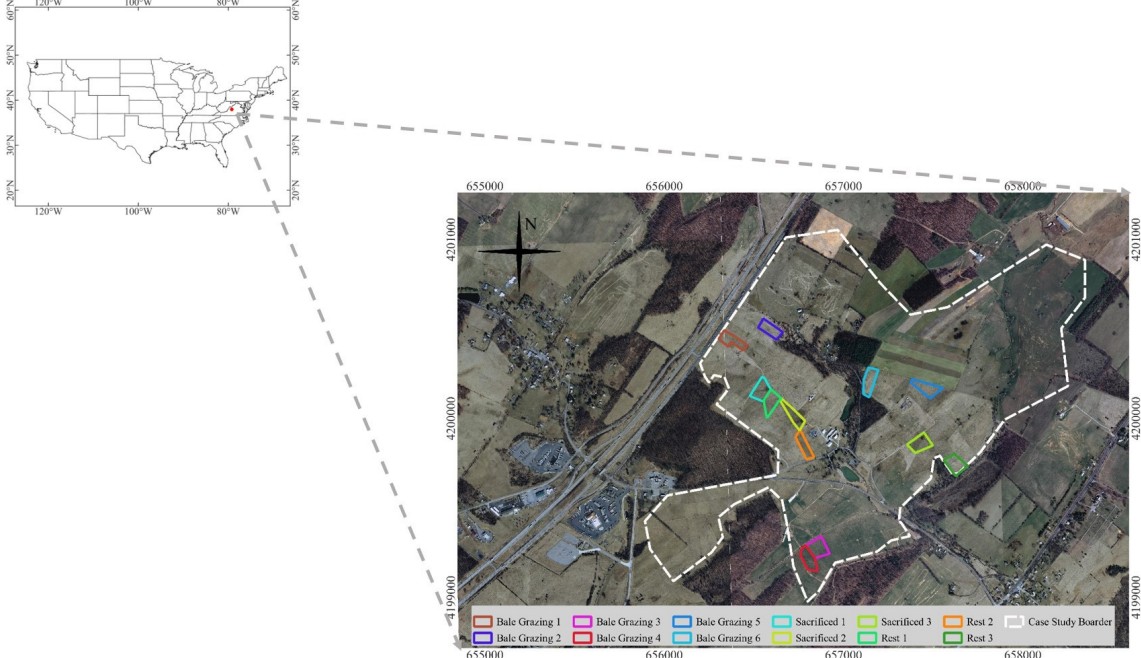

**Figure 1.** An overview of case study border and treatments (paddocks).

### 2.2. Ground Samples

The study involved 12 paddocks with four different treatments: (1) Rotational bale grazing, first paddock grazed; (2) rotational bale grazing, last paddock grazed; (3) sacrifice

paddock; and (4) rest paddock. To ensure rotational bale grazing is not given an advantage or disadvantage, spring forage recovery was evaluated in the first and last paddock grazed in each rotational bale grazing replication. In the rest paddock, paddocks were not grazed throughout the winter and were left to rest while cow/calf pairs fed winter hay in the sacrifice paddock.

Ground forage biomass sampling was conducted over three dates during spring 2022, followed by winter hay feeding. We employed three 0.25 m$^2$ quadrats in each treatment to collect forage biomass samples down to ground level. These samples were collected on three specific dates: 21 April, 27 May, and 14 June. After sample collection, they were carefully placed in a dryer set at 105 °C and left to dry for at least three days until a consistent weight was achieved.

*2.3. UAVs Flights*

Simultaneously with each ground sampling event, we flew a DJI Mavic 2 Pro drone (SZ DJI Technology Co., Ltd., Shenzhen, China) equipped with an RGB camera. The mission planning process was performed using the Pix4Dmapper app 4.8, which is compatible with the drone. We consistently employed the same mission planning strategy for all flights to achieve consistency, accuracy, and comparability in our model. To achieve uniformity, we maintained identical camera settings, altitudes, and trigger intervals for all flights. This approach guaranteed that the captured images exhibited uniform quality and resolution. This level of uniformity is crucial, as variations in camera settings and flight conditions can impact the dependability and accuracy of biomass estimation. The acquired images exhibit high fidelity by upholding uniformity and can be confidently used for precise biomass assessment. These consistent settings also allow the captured images to be compared over time, providing valuable insights into the temporal variability of biomass in the study area.

Given the abovementioned reasons, flights were conducted on 21 April, 27 May, and 14 June using the same mission planning approach outlined in Table 1. The tabulated data indicates consistent attributes across all three flights, including uniform flight altitude, camera settings, flight speed, and side and front overlap configurations. We conducted our flights specifically during local noon on a full sunny day or in an overcast situation.

**Table 1.** The selected parameters in flight mission planning.

| Flight Date | Flight Speed | Flight Altitude | Side Overlap | Front Overlap | GSD (m) |
|---|---|---|---|---|---|
| April | ~3 mph | 60 ft | %75 | %75 | 0.01 |
| May | ~3 mph | 60 ft | %75 | %75 | 0.01 |
| June | ~3 mph | 60 ft | %75 | %75 | 0.01 |

## 3. Methodology

Figure 2 illustrates a step-by-step portrayal of the study's workflow, which consists of five key stages. These stages include selecting research sites and designing experiments, on-site ground sampling, and precise placement of ground control points (GCPs). Following these initial stages, the subsequent phases involve conducting time-series UAV flights across the study area, processing the collected data, and developing models at a dedicated processing station. While the preceding sections provided detailed explanations for the first three steps, the following sections will comprehensively explain the complexities involved in the final two stages.

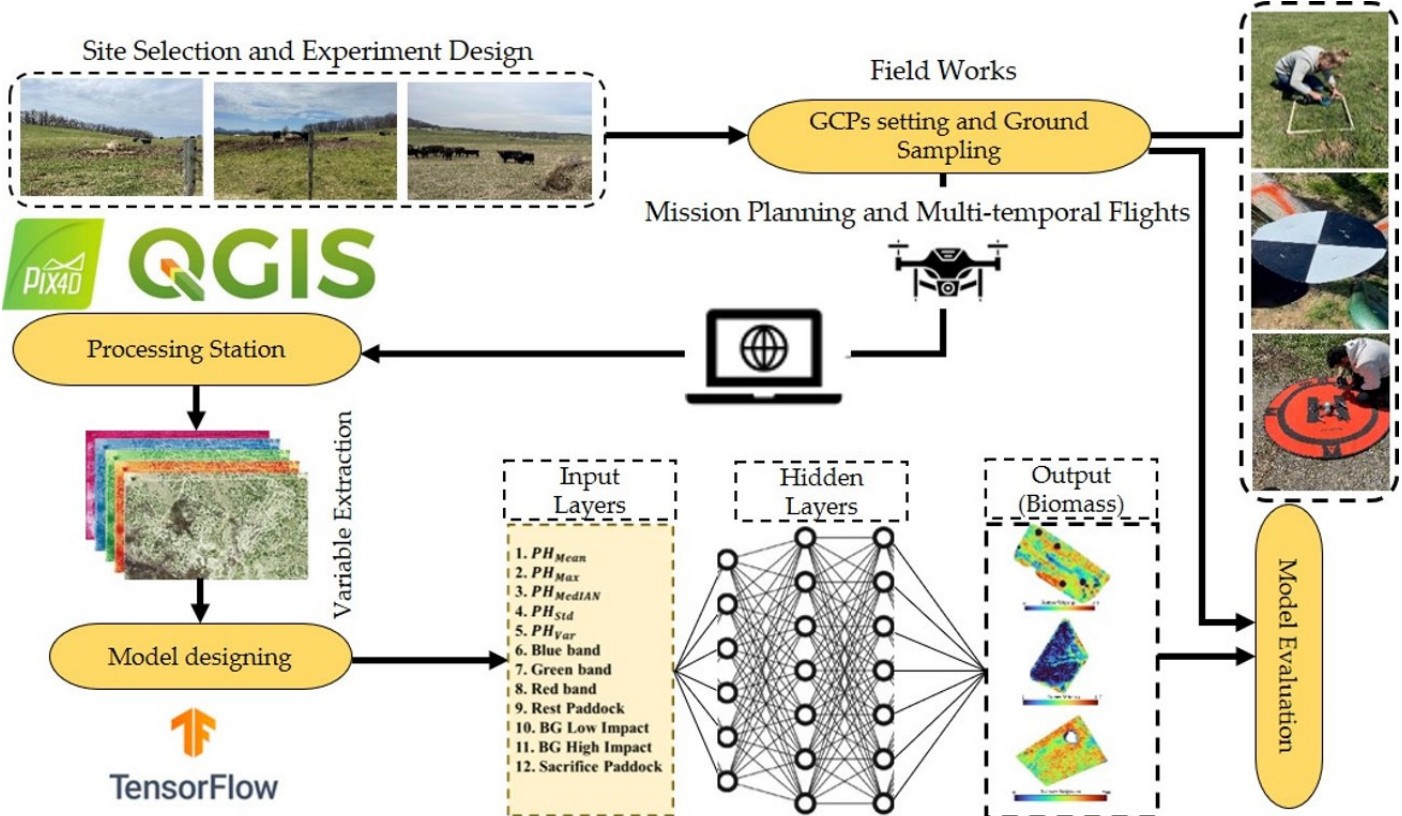

**Figure 2.** The flowchart of the study.

### 3.1. Processing Station

#### 3.1.1. Image Preprocessing

All UAV images were radiometrically and geometrically calibrated using Pix4D software (Pix4D Inc., Prilly, Switzerland). The processing in this software involves generating 3D models and maps from high-resolution images captured by the drone. Some black-and-white pattern targets as GCPs were placed in the corners of each paddock before each flight. The coordinates of the GCP points were measured by Emlid Reach RTK GNSS GPS (Emlid Inc., Saint Petersburg, Russia) units. The GCP points also contributed to the lower number of uncalibrated cameras in the stitching of images. The accuracy of image stitching depends on the quality of the corresponding points found by the software in multiple images. In this study, finding corresponding points in pasture fields with higher biomass and the same pattern (the images captured in June) was challenging in Pix4D. This is because areas with high biomass appear very similar in multiple images, making it difficult for the software to distinguish between them and find corresponding points, resulting in a lower number of calibrated cameras and the accuracy of the final orthomosaic and 3D model. To address this challenge, we relied on the GCP points with superior accuracy in x, y, and z coordinates. Additionally, we meticulously reviewed the report generated by Pix4D software immediately after completing the processing to ensure the accuracy of the georeferencing, orthomosaic, and DSM features. This extra diligence was essential to maintaining the quality and precision of our results.

#### 3.1.2. Image Gridding

Using an object-based decision-making method is consistently recommended when dealing with high-spatial-resolution dataset analysis, which often incurs a substantial computational cost. In this context, two commonly employed techniques, namely image segmentation and image gridding, offer potential solutions for grouping image pixels for classification or regression applications. However, beyond the optimization of computa-

tional efficiency, we were motivated by two additional reasons to develop our model on grids with specific and standardized sizes.

Firstly, the grid system effectively subdivides the pasture into smaller sections or cells, enabling finer-scale biomass quantification. This granularity proves especially crucial in heterogeneous pastures, where biomass significantly varies across the field. By partitioning the pasture into smaller sections, we can capture and account for this variability, ultimately leading to more accurate biomass estimates. Secondly, the gridding approach facilitates the comparison of biomass estimates across different areas within the pasture. This simplifies the identification of regions with high or low biomass and streamlines the extraction of statistical features from the images. These features can serve as input variables in machine learning models for predicting biomass. Therefore, we regarded image gridding as an indispensable step in pasture biomass estimation due to its data organization and analysis efficiency.

We employed QGIS 3.20.2 (QGIS Inc., Lucerne, Switzerland), a free-license software tool for geospatial analysis, to implement the gridding process [29]. This software allowed us to create grids of our desired and standardized size, precisely 1 m × 1 m, which aligns with the dimensions of our ground sampling quadrats. Subsequently, each grid cell was individually selected and subjected to analysis, considering all the features or independent variables detailed in the following section.

### 3.1.3. Variable Extraction

Feature extraction is a pivotal component of regression analysis, serving the essential role of sifting through extensive datasets to pinpoint and transform pertinent information into meaningful attributes [30,31]. These attributes, in turn, become instrumental in forecasting the target variable. The primary objective of feature extraction is to trim the dataset's dimensionality by singling out a subset of the most critical features. This process substantially reduces the computational complexity of the model while elevating its precision [32]. Through this selective extraction of salient variables from the dataset, the model becomes more adept at capturing the underlying relationships between independent and dependent variables, ultimately leading to heightened performance and more precise predictions.

The canopy height model (CHM) is generated by subtracting the digital terrain model (DTM) from the digital surface model (DSM). The DSM represents the elevation of the Earth's surface, including vegetation and man-made structures, while the DTM represents the bare ground elevation without any above-ground features. Mathematically, the CHM is calculated by subtracting DSM from DTM. The resulting CHM represents the height of vegetation or objects above ground level. This process allows us to specifically isolate and quantify the vertical structure of vegetation in the study area, providing valuable information about canopy height for further analysis.

Previous studies have shown that plant height (PH) strongly correlates with vegetation growth and biomass, and incorporating it in the biomass estimation model can enhance the accuracy [16,17,25,33–35]. These studies have also shown that the central tendency (e.g., max, mean) and variability (e.g., standard deviation and variance) of PH are significantly correlated with biomass and less affected by environmental factors such as background soil [36–40]. These metrics, when estimating biomass, could provide an understanding of the overall plant distribution and structure as a representative measure of biomass across the field. As a result, we included these variables in our analysis (Table 2).

In addition, as the plant structures and their heights can impact the plant spectral characteristics [41], we also included mean reflectance values ($Blue_{mean}$, $Green_{mean}$, $Red_{mean}$) of canopy biomass in our analysis. We also included paddock type as a new feature, a categorical feature, in the analysis.

**Table 2.** The defined variables in each grid.

| Variable | Variable Types | Variable Name | Description |
|---|---|---|---|
| Numerical | Structural | $PH_{MAX}$<br>$PH_{Mean}$<br>$P_{Median}$<br>$PH_{Std}$<br>$PH_{Var}$ | Maximum of PH in grids<br>Mean of PH in grids<br>Median PH in grids<br>Standard deviation of PH in grids<br>Variance of PH in grids |
| | Spectral | $Red_{Mean}$<br>$Green_{Mean}$<br>$Blue_{Mean}$ | Mean value of pixels in the red band in grid<br>Mean value of pixels in the green band in grid<br>Mean value of pixels in the blue band in the grid |
| Categorical | Paddock types | BG low impact<br>BG high impact<br>Rest paddock<br>Sacrifice paddock | Low-impact areas in bale grazing<br>High-impact areas in bale grazing<br>Livestock grazing is restricted for a period<br>heavily impacted by trampling |

### 3.2. Model Training

- **Artificial Neural Network (ANN)**

An artificial neural network (ANN) is a machine learning algorithm modeled after the structure and function of the human brain [42]. In this study, the ANN regression model has two hidden layers with 16 nodes each, an activation function of "*relu*", and uses the Adam optimizer.

The mathematical equation for a neural network with two hidden layers can be represented as:

$$Y = f2(f1(X * W1 + b1) * W2 + b2) * W3 + b3 \tag{1}$$

where X is the input data, $W_1$ and $W_2$ are the weights connecting the input to the first and second hidden layer, $b_1$ and $b_2$ are the biases of the first and second hidden layer, $f_1$ is the activation function used in the first hidden layer, $f_2$ is the activation function used in the output layer, $W_3$ is the weight connecting the second hidden layer to the output layer, $b_3$ is the bias of the output layer, and Y is the predicted output.

In this study, the activation function used in the hidden layers was "*Linear*", which means the output of each node in the hidden layer is simply the sum of the inputs multiplied by the weights and added to the bias. The "*Adam*" optimizer is an optimization algorithm that adapts the learning rate during training to speed up convergence and improve accuracy. In this study, we employed the TensorFlow package, implemented in the Python programming language, to create a multi-layer perceptron neural network [43]. We chose TensorFlow over MLPregressor in Scikit-learn due to its greater flexibility and power in deep learning. TensorFlow enables the construction and training of various neural network architectures beyond multi-layer perceptrons (MLPs). With TensorFlow, we could customize several aspects of the neural network, such as the number of hidden layers, the activation function, the initial weight initializer, the optimizer, and the number of neurons in each layer. Using a grid search strategy, we could run multiple artificial neural network (ANN) models with different configurations. All the parameters defined in the ANN model using the grid search strategy are presented in Table 3.

**Table 3.** The inner parameter of ANN, defined in the grid search.

| Epochs | Optimizer | Initializer | Number of Neurons | Activation Function |
|---|---|---|---|---|
| 50, 100, 200, and 500 | "SGD", "RMSprop", "Adam", "Nadam" | "lecun_uniform", "normal", "he_normal", "uniform" | 50, 25, 16, 10, 8, 7, 5, 3 | "Relu", "linear", and "Tanh" |

- *Support Vector Regression (SVR)*

The support vector machine (SVM), introduced by Vapnik in 1995 [44], stands as a robust and potent supervised algorithm employed for addressing both classification and regression (SVR) problems [32,45–48]. The SVM algorithm offers several advantages, including its robust ability to minimize structural risk, proficiency in handling both linear and nonlinear problems, and effectiveness and efficiency, particularly in high-dimensional feature spaces with a limited number of samples. These strengths motivated our choice to implement SVR in this study.

The performance of SVR regression models hinges on the appropriate configuration of inner parameters, specifically the choice of the loss function and the error penalty factor denoted as "C." Additionally, the selection of the kernel function plays a critical role in training the final models. In our study, we concentrated on fine-tuning two crucial parameters: the kernel type and the penalty term (C). To optimize these parameters, we created a parameter grid encompassing various kernel types, including linear, radial basis function (RBF), and polynomial kernels, along with different penalty term values such as 10, 100, and 1000.

- *Random Forest (RF)*

Random forest (RF) regression, introduced by Ho Tin Kam Ho [49], is a machine-learning algorithm employed in both supervised and non-parametric contexts. It operates as an ensemble model composed of multiple decision trees. The use of ensemble techniques, with multiple decision trees making predictions, distinguishes RF from single-model approaches and accounts for its widespread adoption in both regression and classification problems [50]. The primary goal of this algorithm is to construct a "forest" by amalgamating several decision trees, often achieved through the bootstrap aggregation (bagging) method [51]. A key advantage of RF is its ability to combat overfitting issues that may arise from using a large number of features, obviating the need for feature preselection during model training [52]. Additionally, RF possesses two notable advantages over other statistical models: Robustness in the face of noise and the capacity to identify optimal and informative features [52]. RF can deliver strong performance even when confronted with numerous or less valuable features within the input data. The algorithm's performance relies on parameter settings determined by experts during the forest's design or training. Notably, grid search is a valuable technique for optimizing RF models, focusing on two critical parameters: "*n_estimators*" denoting the number of decision trees in the RF ensemble, and "*max_depth*", representing the maximum allowable depth for individual decision trees. To optimize these parameters, we created a parameter grid encompassing various "*n_estimators*" ranging from 20 to 100, along with different "*max_depth*" values such as 5, 7, 8, 10, and 20.

### 3.3. Evaluation Criteria

3.3.1. Common Numerical Criteria

In regression analysis, $R^2$ and RMSE are commonly used evaluation metrics for the model's performance.

$R^2$, also known as the coefficient of determination, is a statistical measure representing the proportion of variance in the dependent variable (Y) that is predictable from the independent variables (X) [53]. It ranges between 0 and 1, with a higher value indicating a better fit of the model. The mathematical equation for $R^2$ is:

$$R^2 = 1 - (\frac{SS_{res}}{SS_{tot}}) \tag{2}$$

where $SS_{res}$ is the sum of squares of the residuals or the differences between the actual and predicted values, and $SS_{tot}$ is the total sum of squares of the dependent variable.

Root mean squared error (RMSE) measures the average magnitude of the residuals or errors between the predicted and actual values. It measures the standard deviation of the

residuals and gives an idea of how close the predicted values are to the actual values [54]. The mathematical equation for the RMSE is:

$$RMSE = \sqrt{\left( \frac{sum\left( \left( Y_{predicted} - Y_{actual} \right)^2 \right)}{N} \right)} \tag{3}$$

where $Y_{predicted}$ is the predicted value, $Y_{actual}$ is the actual value, and N is the number of data points.

In summary, $R^2$ measures the proportion of variance the model explains. At the same time, RMSE measures the model's accuracy by quantifying the differences between the predicted and actual values.

### 3.3.2. Statistics on the Fitted Line between Predicted and Observed Values

In this study, the 1:1 line, referred to as the identity line, was used to compare the agreement between predicted values obtained from the models and observed values. This line plays a pivotal role in assessing the performance of our predictive models. This line of equality serves as a visual benchmark, allowing us to gauge the accuracy and precision of our model's predictions by comparing them to observed values. When our model's predictions align closely with the 1:1 line, it signifies a high level of agreement between the model's outcomes and real-world observations, demonstrating the model's reliability and effectiveness. The presence of data points clustered around this line provides valuable insights into the model's strengths and areas where refinements may be needed. The 1:1 line serves as a crucial tool in the evaluation and validation of our predictive framework, ultimately contributing to the robustness of our research findings.

However, to show the significance of the 1:1 fitted line between predicted and observed values, we used the following statistical criteria. The statistical criteria used to evaluate regression models include the F-statistic [55], which gauges the overall model significance with higher values indicating better fits; the Bayesian information criterion (BIC) [56], which balances model goodness of fit and complexity with lower values favoring parsimony; the *t*-test for slope [57], measuring the significance of relationships between predicted and observed, with larger values indicating stronger relationships; the *t*-test for intercept, assessing the meaningfulness of the intercept; and the probability (Prob) associated with the F-statistic, indicating the model's statistical significance as a whole. These criteria collectively guide the selection of suitable regression models based on their quality and significance.

## 4. Results and Discussion

### 4.1. Variable Analysis

#### 4.1.1. Structural Variables Analysis

This study examined plant and canopy structure changes from April to June 2022, encompassing the transition from low to high biomass density. In each UAV image, for corresponding locations of the ground sampling points, we extracted the max, mean, standard deviation, and variance of plant height, and mean reflectance values in blue, green, and red bands, $PH_{max}$, $PH_{mean}$, $PH_{std}$, and $PH_{var}$, $Blue_{mean}$, $Green_{mean}$, and $Red_{mean}$ respectively. The results are quantified in Table 4 and visually represented in Figure 3. The data in Table 4 highlights a consistent upward trend in average plant height values across all paddocks within the grid. Notably, the increase in average plant height is more pronounced in the rest paddocks compared to the other two. This observation is further supported by the box plots depicted in Figure 3A. In contrast, the average plant height in the sacrifice paddocks lags significantly behind the other two due to the impact of cattle hindering plant growth.

**Table 4.** The time-series statistics calculated for a grid over all the paddocks.

| Variable | HI Bale Grazing | | | LI Bale Grazing | | | Rest | | | Sacrifice | | |
|---|---|---|---|---|---|---|---|---|---|---|---|---|
| | April | May | June | April | May | June | April | May | June | April | May | June |
| $PH_{MAX}$ | 0.084 | 0.16 | 0.71 | 0.206 | 0.255 | 0.84 | 0.35 | 0.7 | 1.11 | 0.106 | 0.25 | 0.536 |
| $PH_{Mean}$ | 0.034 | 0.06 | 0.38 | 0.063 | 0.104 | 0.37 | 0.21 | 0.44 | 0.69 | 0.048 | 0.12 | 0.314 |
| $PH_{Std}$ | 0.017 | 0.04 | 0.11 | 0.040 | 0.056 | 0.14 | 0.052 | 0.1 | 0.11 | 0.020 | 0.05 | 0.086 |
| $PH_{Var}$ | <0.001 | 0.001 | 0.01 | 0.002 | 0.003 | 0.02 | 0.003 | 0.008 | 0.01 | 0.000 | 0.003 | 0.007 |
| $Red_{Mean}$ | 194 | 139 | 143 | 203 | 146 | 122 | 183 | 132 | 129 | 168 | 137 | 141 |
| $Green_{Mean}$ | 184 | 162 | 164 | 193 | 164 | 149 | 197 | 175 | 149 | 171 | 177 | 171 |
| $Blue_{Mean}$ | 159 | 89 | 88 | 182 | 93 | 95 | 146 | 78 | 79 | 126 | 82 | 79 |

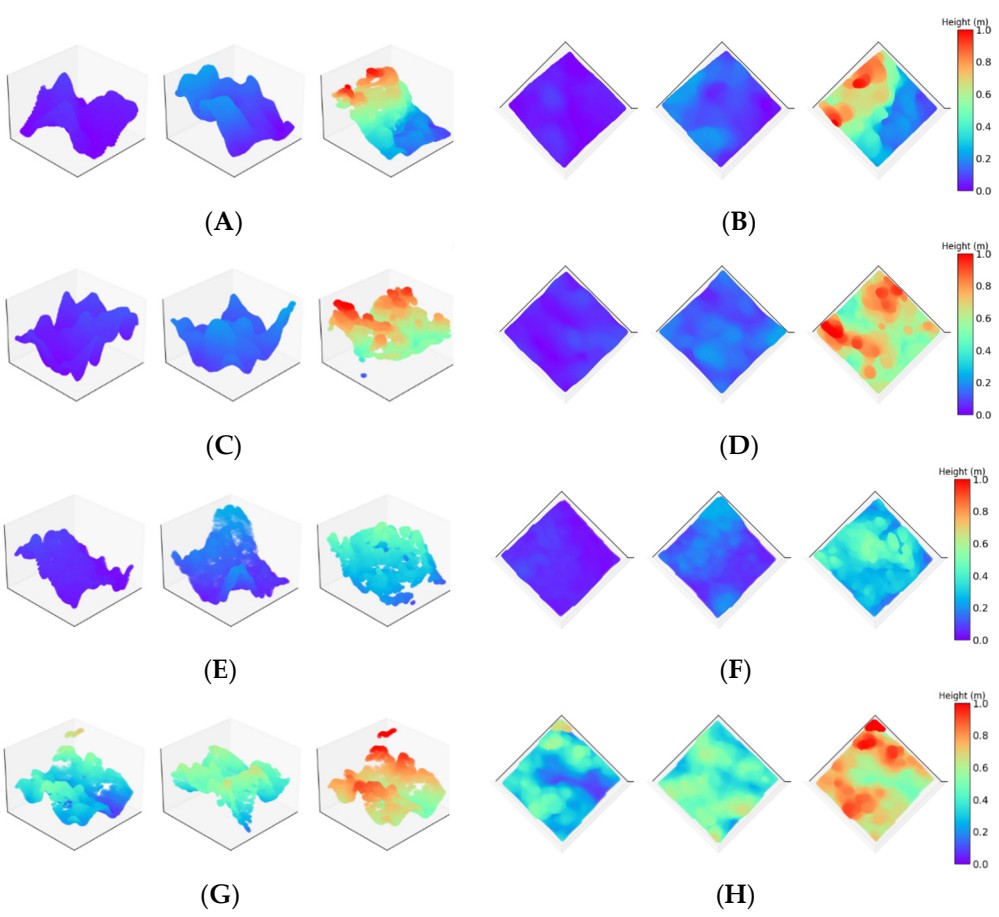

**Figure 3.** The plant height volume (**A**,**C**,**E**,**G**) and surface (**B**,**D**,**F**,**H**) in $1 \times 1 \text{ m}^2$ in high-impact bale grazing, low-impact bale grazing, sacrifice, and rest paddock from April (**left**), May (**middle**) and June (**right**), respectively.

In a broader context, Figure 3 illustrates that in April, plant height in both the Sacrifice and bale grazing paddocks is notably inferior to the rest plots due to the absence of grazing and the ensuing longer plant recovery time. Progressing from April to May, there is a slight increase in average plant height in low-impact bale grazing compared to those with high impact (Figure 3C,D). Interestingly, the sacrifice, bale grazing with low impact, and bale grazing with high impact plots exhibit relatively similar growth (Figure 3A,C,D). Noteworthy differences in average plant height growth materialize from May to June. During this period, the canopy level in rest and bale grazing with low-impact paddocks becomes denser and attains greater height than the other two paddocks (Figure 3C,G).

In the rest paddocks, there is a noticeable increase in plant height during April, attributed to the natural regrowth of vegetation (Figure 3G). This trend continues into May as the plants recover and experience reduced grazing pressure. As the vegetation approaches its seasonal peak in growth, the mean plant height stabilizes and shows slight fluctuations in June (Figure 3G). Figure 3 visually corroborates these findings observed in Table 4, where the average plant height, starting at 0.3 m in April, reaches its zenith at 0.69 m in June.

Notably, the rest paddocks exhibit higher variance and standard deviation in plant height across the grid, indicating more significant variation in the canopy structure compared to the other paddocks. This observation is visually evident in Figure 3G,H), where there is substantial variation in plant height across the grid surface in the rest paddock. In contrast, the canopy structures in the other paddocks appear less varied, with areas of low biomass density or relatively empty spaces.

From April to May, the values of variance and standard deviation of plant height within the grid increase, as reflected in Figure 3, indicating a higher variation in canopy structure compared to April. These two statistics continue to rise from May to June due to increased biomass and plant height variations (Table 4). In contrast, there are no significant changes in the variance and standard deviation values over the pasture canopy in the rest paddock from May to June (Table 4). However, it is noteworthy that plant height obtained from the subtraction of DSM and DTM exhibits a strong correlation of 0.74, indicating a robust linear relationship between this variable and biomass volume. This finding aligns with previous studies that have underscored the reliability and significance of pasture height as an indicator of biomass production [58,59]. In various forage species, including ryegrass, tall fescue, and alfalfa, strong correlations have been reported between pasture height and biomass production [4,60,61].

### 4.1.2. Spectral Variable Analysis

By plotting the changes in spectral reflectance values, the mean values in RGB bands on a grid, we could see the fluctuations in the reflectivity of pasture canopies within the RGB bands throughout the plant's growth cycle, starting from April with minimal biomass and progressing to June with higher biomass (Figure 4). Our ground observations confirmed substantial pasture plant structure, color, and density alterations from April to June as they advance through various growth and developmental stages. Therefore, we sought to find a specific pattern of changes in the canopy structure and spectral reflectance values extracted from the time-series UAV images.

The pasture plants are typically in the early stages of growth in April, at the beginning of the growing season. The plants had lower overall height and limited leaf area, with many still emerging from the soil, proving the higher reflectivity in the blue band and lower values in the height mean (Figure 4B). The density of the plants is relatively sparse, with noticeable gaps between individual plants. The color of the pasture during this period ranges from light green to yellowish as the plants are just starting to photosynthesize and produce chlorophyll, resulting in a decrease in the RGB image values from April to June (Figure 4B–D). These findings corroborate research conducted by Vanamburg et al., (2006) [62], who conducted a comprehensive analysis of grassland color over three months: May, June, and July. Their study demonstrated variations in pasture phenology corresponding to sample dates, with heightened greenness in May, marking the initial stages of plant growth, transitioning to a yellowish hue. Furthermore, they highlighted that shifts in plant phenology and biomass greenness throughout the growth season present challenges in biomass canopy detection.

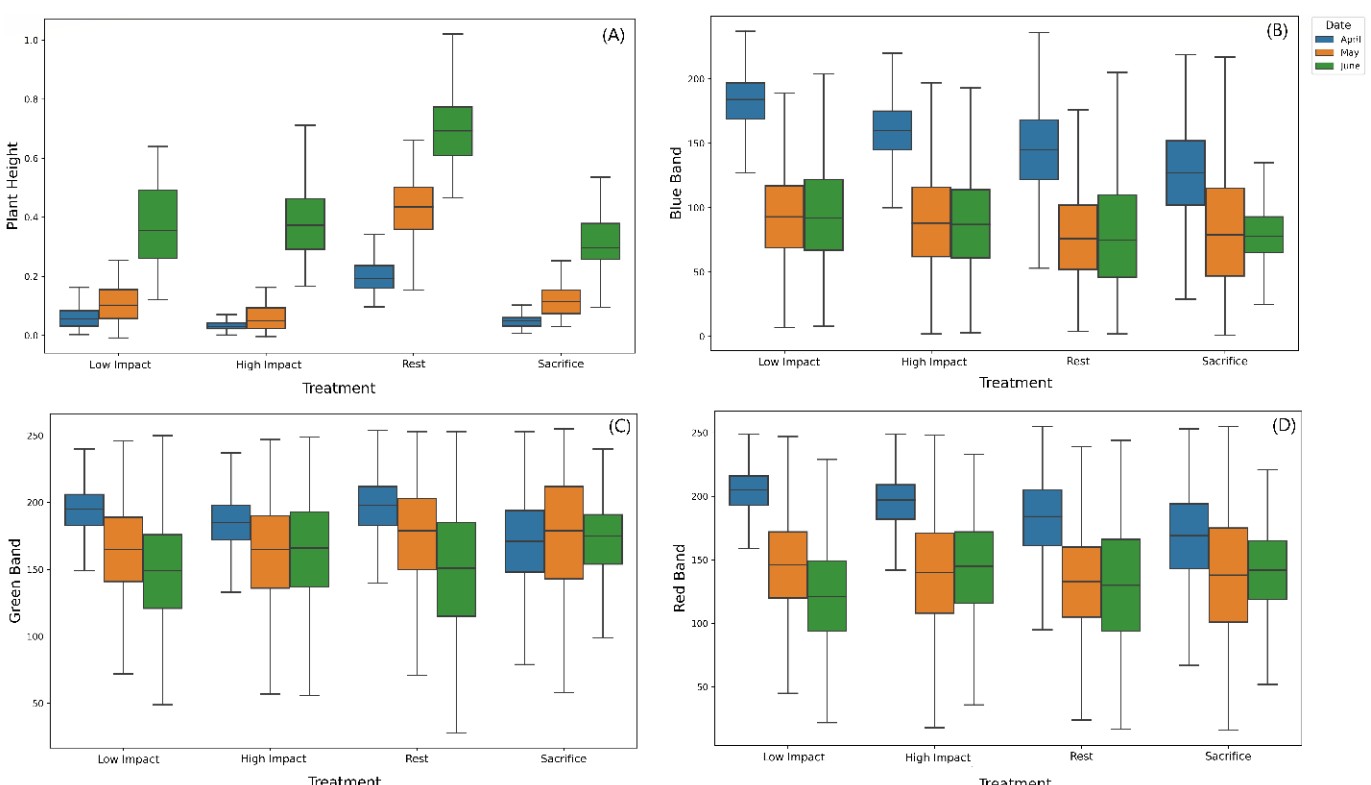

**Figure 4.** The boxplots of variables, showing the variation in RGB spectral band values in paddocks over three months.

Generally, in April, as vegetation awakens from dormancy, the RGB bands might witness a subtle increase due to the initial regrowth and emergence of new leaves. This can result in a slightly elevated greenness (higher green band values) as plants begin photosynthesis and accumulate chlorophyll. The red band might also rise as plants gather momentum in their growth phase. However, our findings show that the canopy structure undergoes notable changes as the season progresses towards June. The plants experience significant growth in terms of height and leaf area (Figure 4B–D). They became more mature and showed a more robust and dense structure. The canopy height increases, with more leaves and stems present, resulting in a denser vegetation cover and lower reflectivity in all the spectral bands (Figure 4B–D). The plants also become more developed in terms of branching and canopy formation, resulting in a more intricate and complex plant structure. There are two possible reasons for decreasing the RGB reflectance values from April to June. The first possible reason is that as biomass increases, the canopy becomes denser and thicker. This leads to increased self-shading within the vegetation, causing less light to reach the sensor. As a result, the recorded reflectance values for the RGB bands decrease. The second reason is the pasture maturity in June, where the pasture undergoes senescence, causing leaves to age and chlorophyll levels to decline. This leads to decreased green and red band reflectance values as the vegetation becomes less vibrant. This finding can also be seen in Figure 5, where there is a diverse relationship between biomass volume and the reflectance in all the red, green, and blue bands. The samples with a lower volume of biomass show higher reflectivity in RGB bands, while a spectral saturation phenomenon occurs for samples with a higher biomass. This phenomenon was mentioned in various studies, and some solutions were also suggested [63–65].

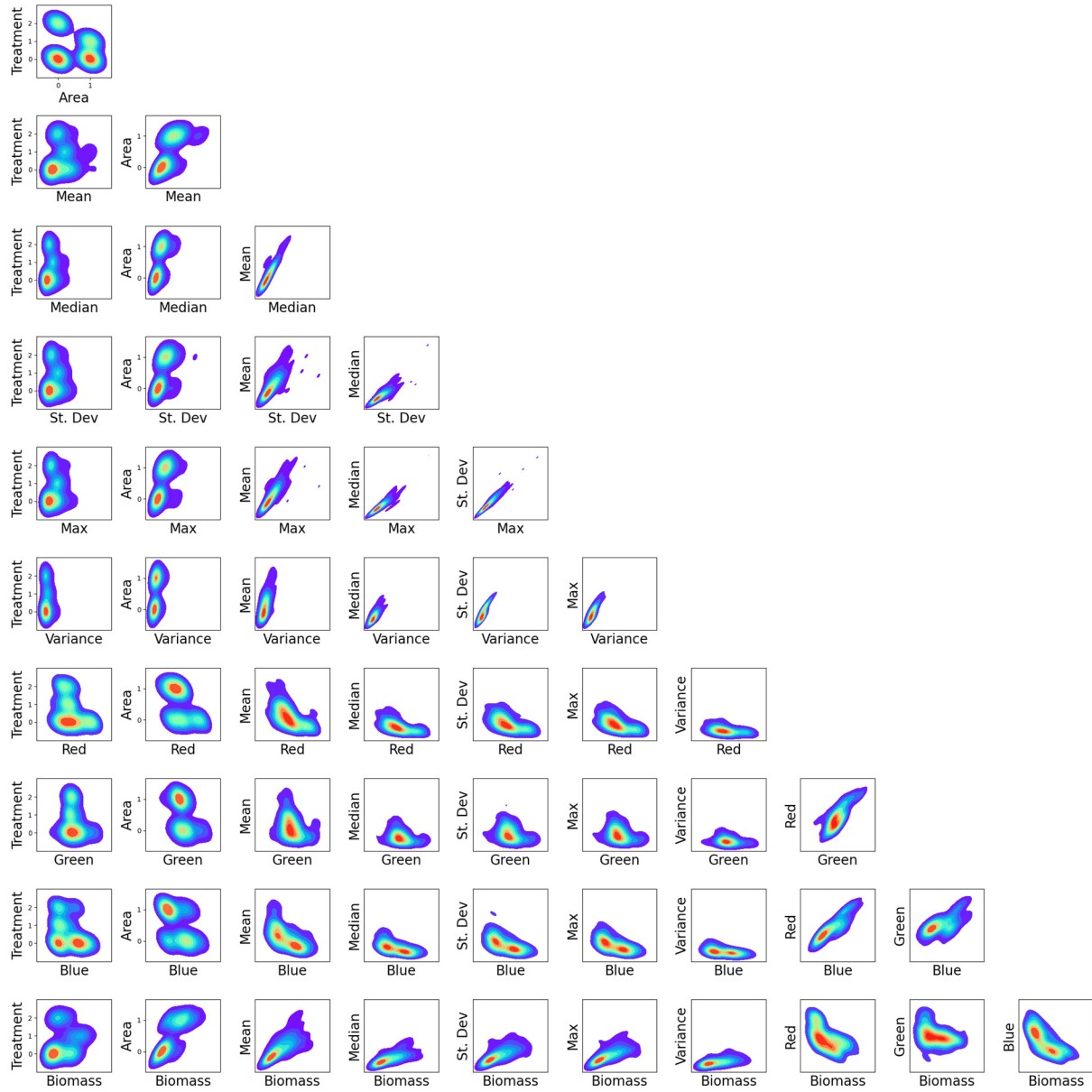

**Figure 5.** The correlation plot of all the independent variables with the biomass variable.

## 4.2. Models Performance

Table 5 presents the performance assessment results for three algorithms, RF, SVM, and ANN, developed in this study. Notably, the ANN model, optimized through a grid search strategy, demonstrates superior performance compared to the other algorithms. Our developed ANN model achieved an estimation of biomass per unit area, boasting an $R^2$ value of approximately 0.93, an RMSE of 62 g/m$^2$, and an MAE of 44 g/m$^2$. The grid search strategy identified optimal ANN model parameters, featuring two hidden layers with 16 neurons each, fully connected, employing the "*Adam*" optimizer and "*relu*" activation function in each layer.

**Table 5.** The evaluation criteria obtained by the optimal ANN, SVR, and RF models.

| Model | $R^2$ | RMSE (g/m$^2$) | MAE (g/m$^2$) |
|-------|-------|----------------|----------------|
| ANN | 0.93 | 62 | 44 |
| SVR | 0.91 | 64 | 48 |
| RF | 0.86 | 78 | 58 |

All implemented models exhibit significant performance, as indicated by $R^2$ in Table 5. However, when comparing the developed ANN with the SVR and RF models in terms of slope and intercept (Figure 6), it becomes evident that the ANN model outperforms the others. Figure 6A illustrates the 1:1 line fitted to the observed vs. estimated biomass, which has a slope of 1.03 and an intercept of 8.08, confirming the superior performance of the ANN compared to the SVR algorithm (slope of 1.05 and intercept of 9.28, Figure 6B). The ANN model also outperforms the RF model with R2, RMSE, and MAE values of 0.86, 78 g/g/m$^2$, and 58 g/g/m$^2$, respectively (Table 5). These results underscore the superiority of the proposed ANN model in quantitative terms when compared with the performance criteria obtained from the SVR model. However, previous studies have reported similar numerical results, with $R^2$ values ranging from 0.45 to 0.98, depending on the dataset used, approaches applied, and extracted independent variables.

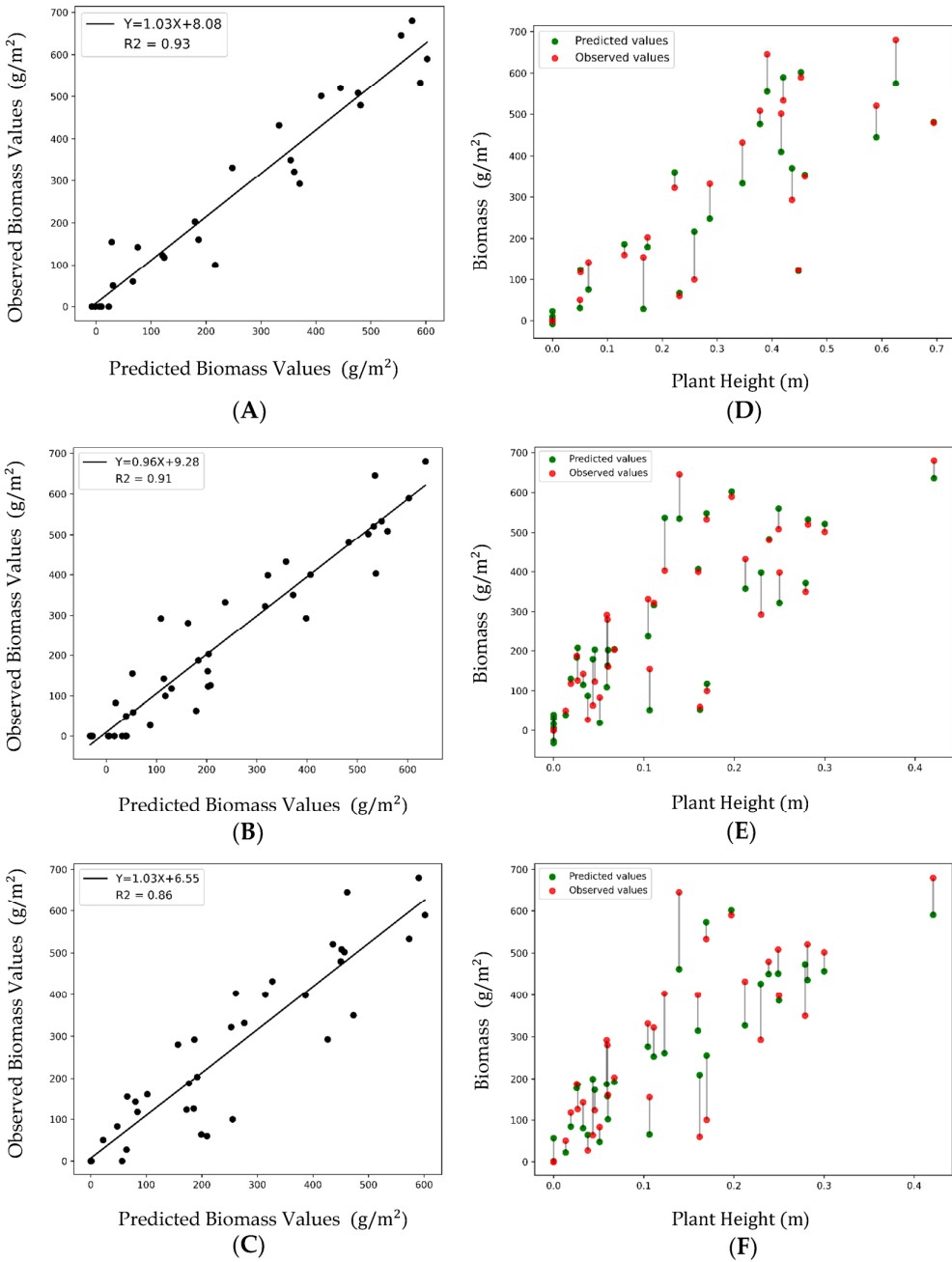

**Figure 6.** The comparison between observed and estimated biomass from ANN, SVM, and RF models ((**A**–**C**), respectively) and the error bar for developed ANN, SVM, and RF models (**D**–**F**).

The better performance of the ANN model can also be seen from the statistical criteria presented in Table 6. The ANN model exhibited the highest F-statistic of 491.4, indicating a strong overall fit. Its Bayesian information criterion (BIC) of 462.4 was the lowest among the models, suggesting good model parsimony. Moreover, the *t*-test for the slope yielded a substantial value of 22.16, emphasizing the significant relationship between predicted and observed values. The *t*-test for the intercept resulted in a value of 1.17, further indicating that the 1:1 line's intercept was significantly different from zero. The exceptionally low probability associated with the F-statistic ($4.43 \times 10^{-24}$) underscores the model's statistical significance, making it a robust choice for predictive modeling.

**Table 6.** The summary of fitted 1:1 line to predicted and observed biomass test samples.

| Model | F-Statistic | BIC | *t*-Test/Slope | *t*-Test/INTERCEPT | Prob (F-Statistic) |
|-------|-------------|-----|----------------|---------------------|---------------------|
| ANN | 491.4 | 462.4 | 22.16 | 1.17 | $4.43 \times 10^{-24}$ |
| SVR | 390.3 | 471.3 | 19.75 | 0.62 | $3.05 \times 10^{-22}$ |
| RF | 256.1 | 487.5 | 16.03 | 0.014 | $5.50 \times 10^{-19}$ |

The SVR model also demonstrated strong performance, with a respectable F-statistic of 390.3, suggesting a significant overall fit. However, the BIC value was slightly higher at 471.3 compared to ANN, indicating a relatively less parsimonious model. Nevertheless, the *t*-test for the slope, with a value of 19.75, highlighted a weaker performance compared to the ANN model. The *t*-test for the intercept yielded a value of 0.62, which was statistically different from zero. The probability associated with the F-statistic ($3.05 \times 10^{-22}$) confirmed the model's overall significance, making it a viable alternative for regression analysis.

On the other hand, the RF model, while still performing well, exhibited a lower F-statistic of 256.1 compared to ANN and SVR, indicating a slightly weaker overall fit. Its BIC value of 487.5 was the highest among the models, implying a less parsimonious model structure. However, the *t*-test for the slope resulted in a substantial value of 16.03, suggesting a significant relationship between the variables. Surprisingly, the *t*-test for the intercept produced a value of 0.014, which, although different from zero, was the lowest among the models. The probability associated with the F-statistic ($5.50 \times 10^{-19}$) reinforced the model's significance, making it a reasonable choice for regression tasks.

However, the ANN model exhibited the highest overall performance, as indicated by the F-statistic, BIC, and *t*-tests for both the slope and intercept. The SVR model followed closely, offering a strong fit and significance. While the RF model showed slightly lower overall performance.

The superior performance of ANN in our biomass estimation study is attributed to its ability to learn and model complex, non-linear relationships between spectral bands and biomass, especially in the presence of spectral saturation. The power of artificial neural networks (ANN) in handling non-linearity and complex relationships becomes particularly beneficial when the ANN model can show high adaptivity and capture intricate non-linear relationships between input variables (spectral bands) and the target variable (biomass). This means that ANNs can learn to model the saturation effect and understand that there is not a straightforward linear relationship between spectral bands and biomass in situations where saturation occurs. From the spectral analysis part of this study, given in Figure 5, it could be seen that biomass estimation using spectral bands faces spectral saturation phenomena in which spectral bands may saturate and even decrease as biomass increases. The ANN model could profoundly deal with this issue and could show better performance in biomass estimation, regarding the presented numerical evaluation criteria in Table 6.

Moreover, feature extraction and representation learning are another advantage of the ANN model that proved its performance over the other two models. The ANN has the capacity to learn hierarchical representations from the data. In cases of spectral saturation, the ANN model could learn to extract relevant features or transformations of the spectral data that better capture the underlying relationship with biomass. This feature extraction

can help the model handle situations where the direct spectral values may not provide a clear linear relationship.

### 4.3. Comparison with Previous Studies

To facilitate a fair comparison of our study's algorithm performance with that of previous studies in terms of achieved $R^2$ values, we should emphasize the evaluation criteria employed by similar studies in terms of the used dataset (RGB images) and targeted dependent variable to be estimated (pasture biomass). In addition, the accuracy of estimation may vary depending on the implemented model, either a learning tool or a parametric model, and the independent variables utilized in the models (Table 7).

**Table 7.** The recent studies conducted on pasture biomass estimation using UAV-RGB images.

| Author | Year | Spectral Info | Structural Info | Model | $R^2$ | Ref |
|---|---|---|---|---|---|---|
| Batistoti et al. | 2019 | n/a | Plant height | Linear regression | 0.74 | [66] |
| Borra-Serrano et al. | 2019 | 10 spectral indices | 7 canopy metrics | PLSAR, MLR, LR and RF | 0.67, 0.81, 0.58, 0.70 | [67] |
| Castro et al. | 2020 | 4 spectral indices | n/a | CNN | 0.88 | [68] |
| DiMaggio et al. | 2020 | n/a | Plant height | LR | 0.65 | [69] |
| Grüner, et al. | 2019 | n/a | Plant height | LR | 0.72 | [17] |
| Lussem et al., | 2019 | 6 spectral indices | Plant height | MLR | 0.73 | [70] |
| Lussem, et al. | 2020 | n/a | Plant height metrics | LR | 0.86 | [16] |
| Alves Oliveira et al. | 2022 | n/a | n/a | CNN | 0.79 | [71] |
| Qin et al. | 2021 | Spectral indices | Fractional vegetation cover | LR | 0.45 | [27] |
| Rueda-Ayala et al. | 2019 | n/a | Plant height metrics | LR | 0.54 | [37] |
| Shorten and Trolove | 2022 | Mean spectral bands for vegetative and soil material | Percent vegetation cover and forage volume | LR | 0.66 | [72] |
| Sinde-González et al. | 2021 | n/a | Density factor and volume | Descriptive statistic | 0.78 | [11] |
| Van Der Merwe, Baldwin and Boyer | 2020 | n/a | Canopy height model | LR | 0.91 | [33] |
| Vogel et al. | 2019 | Reflectance of red, green, and blue; hue: saturation, value | n/a | LR | 0.81 | [73] |
| Wijesingha et al. | 2019 | n/a | 10 canopy height metrics | LR | 0.62 | [74] |
| Zhang et al. | 2022 | 6 color space indices and 3 vegetation indices | Canopy height model from point clouds | RF | 0.78 | [22] |

As presented in Table 7, the reported coefficients of determination vary across studies, ranging from 0.45 in a study by Qin et al., where they employed both spectral and fractional vegetation cover as structural variables in a linear model. This indicates that, despite considering both structural and spectral variables, accurate pasture biomass estimation may necessitate the use of complex models capable of handling high-dimensional variable space, and considering all the potential spectral and structural independent variables into a model cannot guarantee an accurate pasture biomass estimation.

The maximum coefficients of determination, with a value of 0.88, were reported by Castro et al., who refrained from extracting any variables from the RGB images and instead trained a CNN model for biomass estimation. They examined the performance of CNN deep neural networks designed by different architectures (AlexNet and ResNet), and the reported accuracy ($R^2$ = 0.88) was the highest, which is still lower than the reported $R^2$ obtained from the ANN model in this study, where we analyzed the variation of pasture's structure and phenology behavior over a specific period of time. As a result, a complicated model cannot still achieve desirable accuracy in biomass estimation, and an accurate pasture biomass estimation needs a model trained by significant variables. The variables, either structural or spectral, are extracted from the images, coming from experts' knowledge of pasture structure and phenology.

Interestingly, studies that use both structural and spectral variables tend to yield better estimates of pasture biomass. However, despite this advantage, their reported accuracy ($R^2$) falls short of the achieved $R^2$ in our study, underscoring the significance of employing machine learning tools, specifically the ANN model, which outperforms both SVR and RF models.

In comparison to a prior study utilizing 3D information to estimate biomass for the same species, our research yields more precise results. Batistoti et al. obtained ($R^2$ value of 0.74, while another study by [75] focused on estimating yield in heterogeneous grassland swards using proximal sensing equipment and the multiple partial least square regression (MLPSR) approach, achieving an $R^2$ of 0.69. The improved $R^2$ value in our study may be attributed to the capability of machine-learning algorithms to model nonlinear relationships between independent and dependent variables, surpassing the MLPSR approach. Other studies that used machine learning algorithms, such as Borra et al. and Zhang et al., employing RF regression, demonstrated promising accuracy. The RF model developed in our study, with an $R^2$ of 0.86, outperforms the RF model developed by Zhang et al. in 2022, which achieved an $R^2$ of 0.78. This enhanced performance can be attributed to the experimental design of our study, which considered different paddock types and involved time-series flights over the paddocks, providing a profound variation in pasture biomass.

The merits of UAV RGB images were also highlighted in a study where machine learning techniques were employed for grassland biomass estimation from spectral data, resulting in a root mean square error (RMSE) of 71.2 t·ha$^{-1}$. These findings underscore the capability of our RGB dataset collected using UAVs, coupled with robust methods like deep neural networks, to compete with costly remote measurement systems and even offer advantages over terrestrial measurements. Additionally, Danyang et al. compared RF, SVM, and MLR performance in maize crop aboveground biomass estimation. Their results, combining RGB images with either SVM or RF, were capable of estimating biomass with an $R^2$ of 0.90 and 0.89, respectively. These results closely resemble the numerical outcomes of our model, further emphasizing the superior performance of our ANN model over the RF and SVM models. An intriguing observation from previous studies is the prevalence of linear regression and PLSAR algorithms in estimation work. Linear regression was frequently used, even though we have demonstrated that ANN holds great potential for biomass estimation.

### 4.4. Biomass Maps

The primary goal of this study was to create maps displaying biomass quantities within a specific area (measured in grams per square meter, g/m$^2$). Figures 7–9 illustrate a series of biomass maps generated using our ANN model spanning from April to June, covering three distinct paddock types: Bale grazing, rest, and sacrifice, respectively. In the biomass maps of bale grazing paddocks, it is evident that the regions where bales were placed (indicated by circles in the paddocks) exhibited lower biomass levels over the three months. This outcome aligns with our on-site field observations. These areas, referred to as high-impact zones, are influenced by cattle activity and consequently experienced only modest biomass growth during the three-month period. Nevertheless, there was a

noticeable increase in biomass volume in the bale grazing paddocks, particularly from May to June, almost reaching the biomass levels observed in the rest of the paddocks (Figure 8). This growth pattern demonstrates that the developed ANN model effectively captures changes in biomass maps over the three-month period, with a particular emphasis on estimating lower biomass quantities in the bale-placed areas.

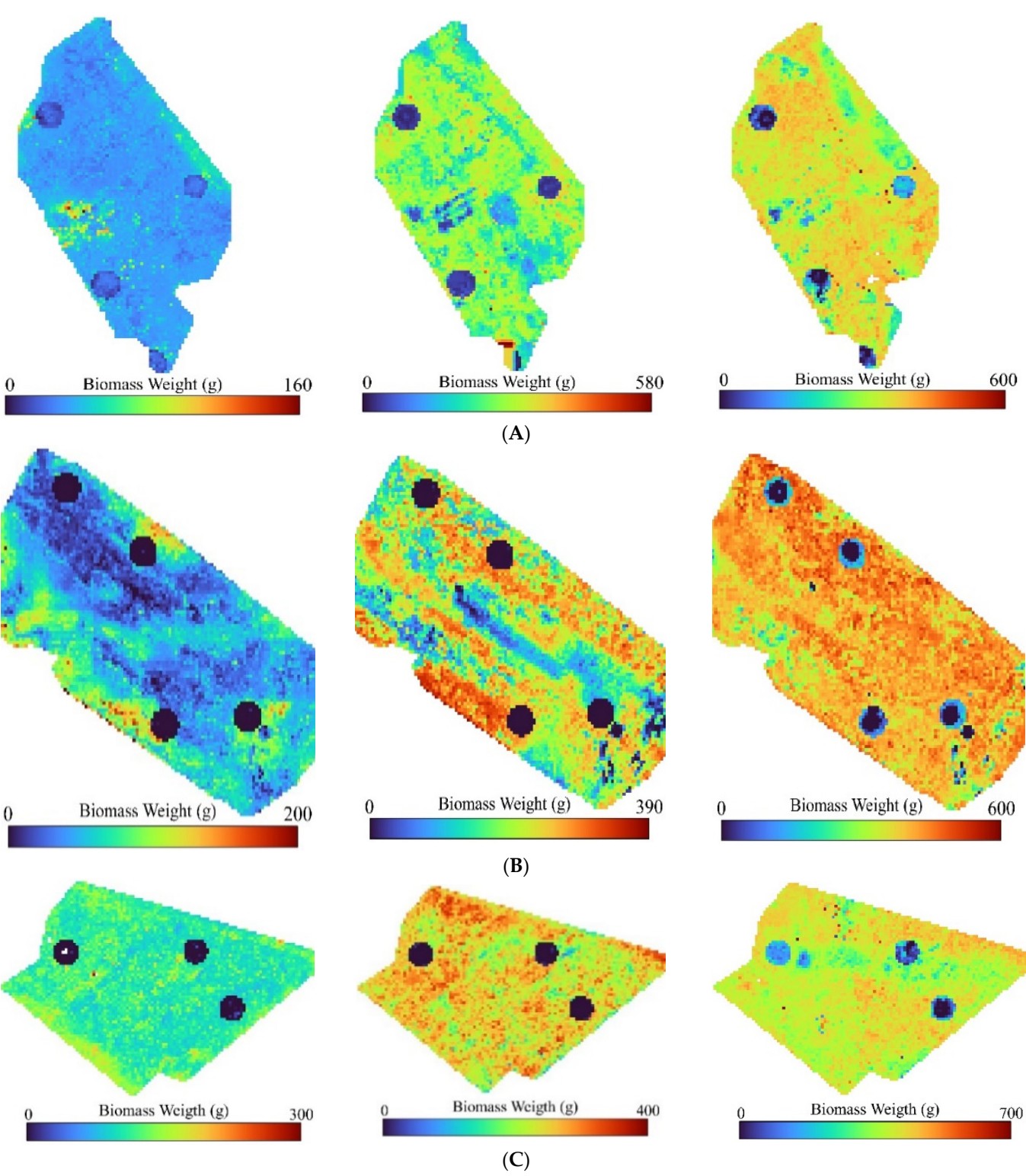

**Figure 7.** The biomass maps generated by the ANN model in three bale grazing paddocks over April (**left**), May (**middle**), and June (**right**) month, (**A–C**) stand in paddocks with a different name).

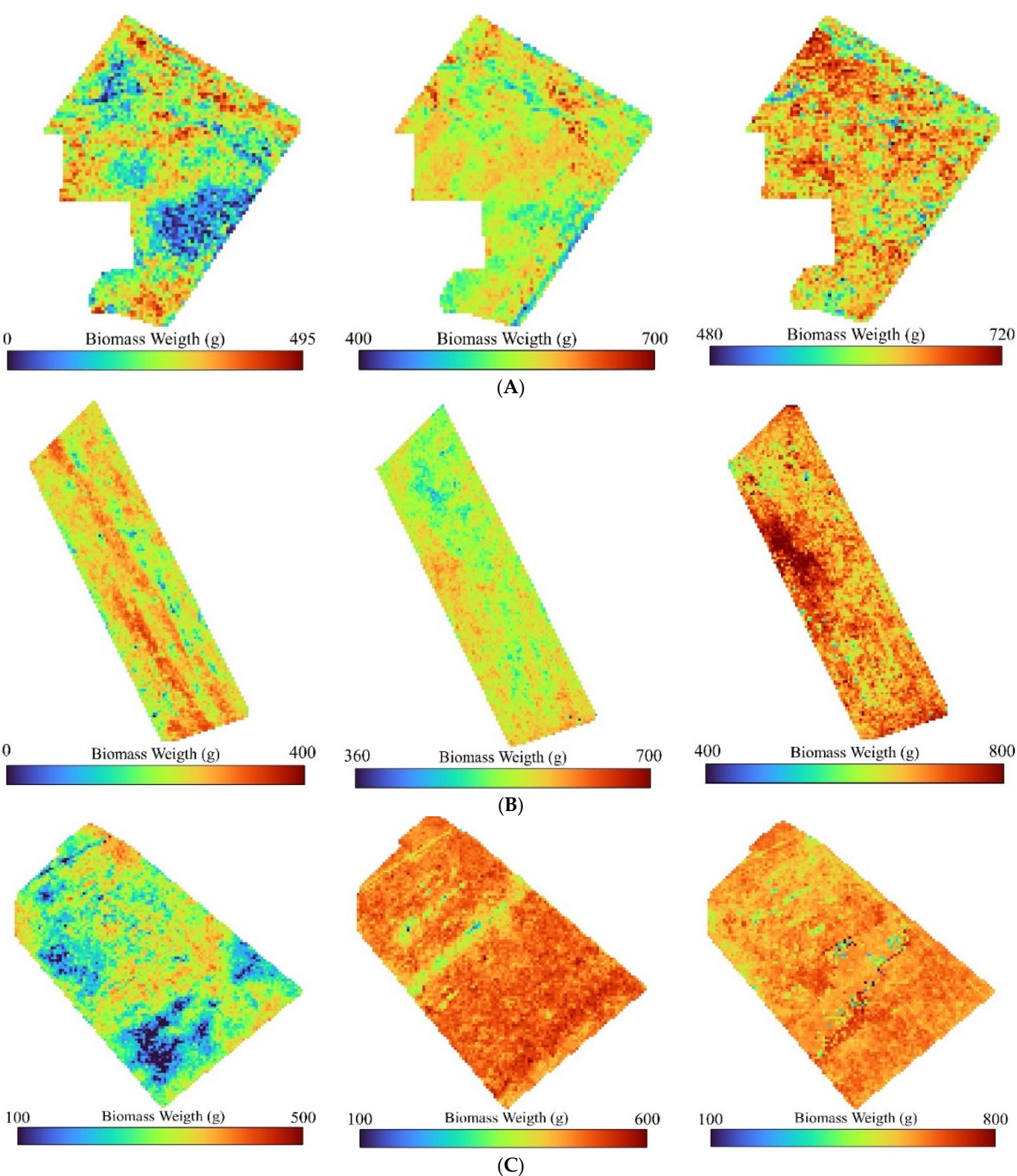

**Figure 8.** The biomass maps generated by the ANN model in three rest paddocks over April (**left**), May (**middle**), and June (**right**) month, (**A–C**) stand in the paddocks with a different name.

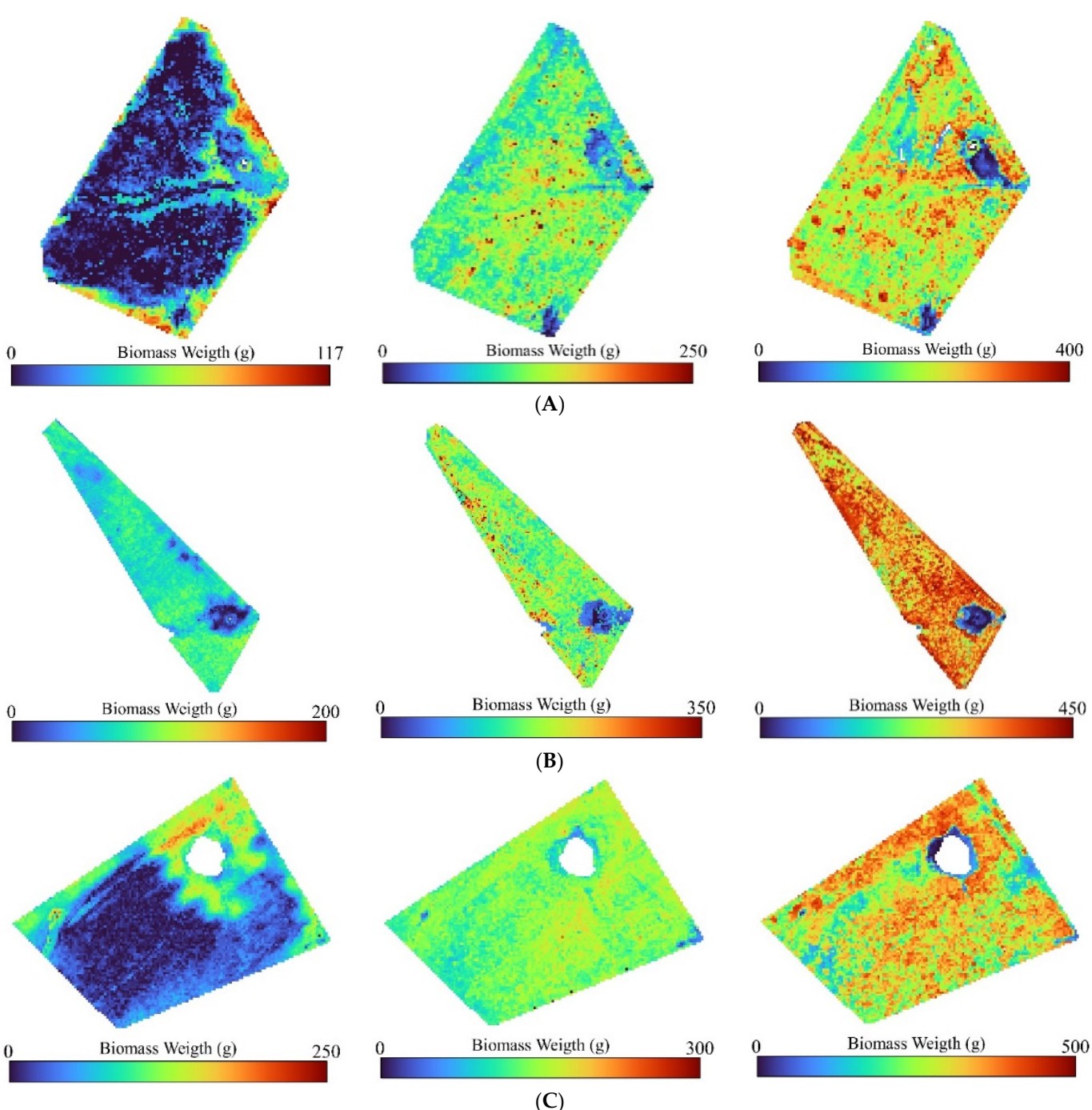

**Figure 9.** The biomass maps generated by the ANN model in three sacrifice paddocks over April (**left**), May (**middle**), and June (**right**) month, (**A–C**) stand in paddocks with a different name).

Figure 8 presents biomass maps for three rest paddocks over three months of April, May, and June (left, middle, and right figures, respectively). Based on our ground-level observations, rest paddocks generally exhibited slightly higher biomass volumes, as they were not subject to grazing activities during the three-month period of our study. The ANN model results mirror this biomass pattern, showing that in nearly all paddocks classified as "rest", the biomass volumes were higher compared to the other two paddock types. This underscores the significance of including paddock type as an independent variable in the model, as it contributes to the production of more dependable biomass maps.

Figure 9 provides a comprehensive visualization of biomass distribution maps for three "sacrifice paddocks" spanning three consecutive months: April, May, and June (depicted

in the left, middle, and right figures, respectively). Sacrifice paddocks are recognized as high-impact areas due to the extensive trampling and disturbance caused by cattle grazing. Based on our ground-level observations, it was evident that, across all three sacrifice paddocks, almost in all three time-series ground sampling, the biomass weight was consistently lower when compared to both "Bale Grazing" with low and high impact areas and rotational grazing and "Rest" paddocks with low impact area and no grazing. Rest paddocks, characterized by low impact and no grazing during the growing season, exhibited significantly higher biomass throughout the entire growing season in comparison to sacrifice paddocks, which experienced greater surface impact.

By visually comparing the biomass maps presented in Figures 7–9 with the legend indicating the maximum and minimum biomass weights, a consistent pattern in biomass distribution becomes evident. This observation underscores the significance of incorporating paddock types as a categorical variable within the artificial neural network (ANN) model. Such inclusion enhances the model's capacity to make more accurate estimations that closely align with real-world conditions and ground-based samples. This refined modeling approach not only accounts for spatial variations in biomass but also considers the distinctive characteristics and management practices associated with different paddock types, thus contributing to more precise biomass predictions. Consequently, the integration of paddock types as a categorical feature bolsters the model's robustness and improves its ability to capture the complex dynamics of biomass distribution in diverse agricultural settings.

## 5. Conclusions

In this study, we conducted a comprehensive time-series analysis encompassing both RGB image values and the structural characteristics of pasture canopies across three predominant paddock types throughout an entire plant growth cycle. Subsequently, we harnessed the dynamic variations in statistical variables obtained from the CHM and RGB image values as input vectors for an ANN model. We aimed to accurately estimate pasture biomass within standardized 1 $m^2$ units. We achieved a holistic understanding of pasture canopy dynamics by integrating temporal analyses of structural and color-related parameters. CHM statistics shed light on the progressive transformation in canopy structure and biomass density over the plant growth period, while RGB image values provided supplementary insights into evolving plant phenology. These diverse datasets enriched our comprehension of pasture dynamics and facilitated the development of an effective biomass estimation model.

The developed ANN model underwent rigorous evaluation, incorporating statistical criteria and ground observations. This comprehensive assessment underscored the model's remarkable accuracy in estimating biomass volume, yielding an $R^2$ value of 0.94 and an RMSE of 62 ($g/m^2$). These numerical evaluations underscored the pivotal role played by ultra-high-resolution photogrammetric CHMs and red, green, and blue (RGB) values in capturing meaningful variations, significantly enhancing the model's precision across various paddock types, including bale grazing, rest, and sacrifice paddocks. Furthermore, the generated maps visually show the model's sensitivity to areas with minimal or negligible biomass during the plant growth period. Notably, it effectively identified low-biomass areas in bale grazing paddocks and regions with reduced biomass impact in sacrifice paddocks compared to other types. These maps underscore the model's versatile capacity to estimate biomass across diverse scenarios, ranging from locations with minimal biomass to those with substantial accumulation, confirming its adaptability for deployment across varying paddock types and conditions.

In our forthcoming research phase, we will pursue three primary objectives. Firstly, we intend to scrutinize the efficacy of high-resolution RGB imagery in biomass estimation while considering various flight parameters, such as altitude, speed, and overlaps, which will provide valuable insights for optimizing flight precision. Secondly, we aim to assess the model's performance and the utility of UAV RGB imagery for biomass es-

timation across a spectrum of pasture types, including cold-season pastures. Lastly, we will compare drone-based sensors, including hyperspectral and multispectral imaging, alongside high-resolution RGB imagery to assess their respective roles in accurate pasture biomass estimation. This analysis will shed light on their unique capabilities within pasture environments.

**Author Contributions:** Each author made substantial contributions to this publication; M.V. and S.T. carried out the experimental work and collected the data; S.S., M.V. and R.M. analyzed the data and wrote the manuscript. All authors have read and agreed to the published version of the manuscript.

**Funding:** This research received no external funding.

**Data Availability Statement:** No new data were created or analyzed in this study. Data sharing is not applicable to this article.

**Conflicts of Interest:** The authors declare no conflict of interest.

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
