# Peer review of "Pasture Biomass Estimation Using Ultra-High-Resolution RGB UAVs Images and Deep Learning"

_remotesensing, doi:10.3390/rs15245714_

Round 1

Reviewer 1 Report

Comments and Suggestions for Authors

This is an interesting article which falls within the scope of Remote Sensing and the intended SI (UAS Technology and Applications in Precision Agriculture). I do have some comments (mainly about the description of the methods), which should be addressed before reconsidering this article for publication.

- Abstract / L-16, L 19, L23: This needs to be restructured: the abbreviation (e.g. CHM on L16) is mentioned before it is explained (e.g. Canopy Height Model on L23). 

- L37 - L46: Please add references underpinning these important statements.

- L47 - L56: To improve the interpretation of this section (and later on the results) it might be good to add the spatial resolution of the satellite imagery used in these studies. This can be revisited in the discussion of the ultra high-resolution UAV imagery.

- Table 1: This a rather low flight altitude (60 ft). Did the authors consider a higher altitude since this would have resulted in less images and a lower computational capacity? Or is this due to drone legislation? In contrast, the authors opted for a rather low image overlap (75%), which is probably the minimum overlap that is necessary to accurately stitch the images (and perform structure-from-motion). It would be interesting if the authors can give an overview of the amount of images that were collected per site.

- Section 3.1.2 (line numbers seem to be missing here): which version of QGIS was used? And can you further explain the rationale behind the image gridding at 1m² and the dimensions of the ground sampling quadrats (0.25 m², as mentioned in L142)

- L207: I assume that plant height is computed based on the CHM (= DSM-DTM). This step seems to be missing and should be explained here (or in 3.1.1). In addition, I wonder how accurate the DTM (and thus CHM) based on the summer imagery is, as the ground level is hard to detect when vegetation is abundant. If this was an issue, it can be recommended to use the DTM of the period in which the PHmean was the lowest (i.e. April based on Table 4).

- Discussion: I think that this research still requires some additional discussion. Please compare your results (incl. research design) with those from other studies. It might be interesting to reflect on the implementation of other sensors (e.g. LiDAR), and on the fusion of UAV and satellite imagery (and the other suggestions you mentioned in the last part of the Conclusion).

Author Response

Dear Reviewer,

We are grateful for your valuable feedback and thoughtful comments on our manuscript titled "Pasture Biomass Estimation Using Ultra High-Resolution RGB UAVs Images and Deep Learning." Your insights are greatly appreciated, and we have addressed your comments and suggestions as follows:

  1. Abstract / L-16, L 19, L23: This needs to be restructured: the abbreviation (e.g. CHM on L16) is mentioned before it is explained (e.g. Canopy Height Model on L23). 

Thanks a lot for your comment. We corrected it based on your comment.

  1. L37 - L46: Please add references underpinning these important statements.

Thanks a lot for your comment. We added three references for that part.

  1. L47 - L56: To improve the interpretation of this section (and later on the results) it might be good to add the spatial resolution of the satellite imagery used in these studies. This can be revisited in the discussion of the ultra high-resolution UAV imagery.

Thanks a lot for your comment. We added the spatial resolution for each study in that part.

  1. Table 1: This a rather low flight altitude (60 ft). Did the authors consider a higher altitude since this would have resulted in less images and a lower computational capacity? Or is this due to drone legislation? In contrast, the authors opted for a rather low image overlap (75%), which is probably the minimum overlap that is necessary to accurately stitch the images (and perform structure-from-motion). It would be interesting if the authors can give an overview of the amount of images that were collected per site.

Regarding the choice of a low flight altitude (60 ft) and the 75% image overlap in our study, we appreciate the reviewer's insightful comment. The decision to fly at this altitude and maintain a 75% image overlap was indeed influenced by several factors, including drone capabilities and environmental conditions.

Drone Capability: We used a small Mavic Pro drone for our data collection, which has certain limitations, particularly in windy conditions. Attempting to fly at a higher altitude in a windy area could lead to stability and safety issues, as indicated by the drone's error messages. Therefore, to ensure the safety and stability of the data collection process, we had to make the trade-off of flying at a lower altitude.

Image Overlap: The 75% image overlap was selected as it is generally considered the minimum necessary for accurate image stitching and structure-from-motion processing. This overlap ensures that the software can effectively match and align the images to create accurate 3D models. While a higher overlap might be desirable in ideal conditions, it would have resulted in a significantly larger number of images, increasing the computational requirements and processing time. Given our constraints, the chosen overlap provided a balance between data quality and practical considerations.

Concerning the quantity of images obtained, we collected a total of 4,440 images per flight, each flight resulting in a data storage requirement of 72 Gb

In summary, the choice of a low flight altitude and 75% image overlap was influenced by both drone capabilities and the need to ensure accurate data collection under the specific environmental conditions of the study area. We believe that these considerations were essential for maintaining data quality and safety during the data acquisition process.

  1. Section 3.1.2 (line numbers seem to be missing here): which version of QGIS was used? And can you further explain the rationale behind the image gridding at 1m² and the dimensions of the ground sampling quadrats (0.25 m², as mentioned in L142)

Thanks a lot for your comment. The QGIS version is 3.20.2. We added it to the manuscript.  We appreciate the reviewer's inquiry regarding the rationale behind the chosen image gridding resolution at 1m² and the ground sampling quadrat dimensions of 0.25 m² (as mentioned in Line 142). In the existing body of literature, numerous investigations have employed a ground sampling size of 0.25m² as a widely adopted measure for forage cutting. Nevertheless, our aim was to identify a standardized and optimal size that could effectively streamline the computational processes involved in map generation while preserving all pertinent details. The decision to use a 1m² image gridding resolution was driven by the need to strike a balance between capturing detailed information about the study area and maintaining computational efficiency. A finer resolution may result in an excessively large dataset, posing challenges in terms of computational resources and processing time. On the other hand, a coarser resolution might lead to a loss of critical spatial information. The 1m² resolution was deemed optimal for our study as it allowed us to sufficiently capture relevant spatial patterns and features while managing computational demands effectively.

  1. L207: I assume that plant height is computed based on the CHM (= DSM-DTM). This step seems to be missing and should be explained here (or in 3.1.1). In addition, I wonder how accurate the DTM (and thus CHM) based on the summer imagery is, as the ground level is hard to detect when vegetation is abundant. If this was an issue, it can be recommended to use the DTM of the period in which the PHmean was the lowest (i.e. April based on Table 4).

Thank you for the comment! The explanation has been added to the section 3.1. 3..  Regarding the accuracy of the DTM, we understand the potential challenges posed by dense vegetation during the summer period. Your suggestion to consider the DTM from a period when the mean plant height is lowest, such as April (as indicated in Table 4), is insightful. We actually used the DTM generated from the first flight (April) as there was very low amount of biomass and ground surface could be seen from some spots over the paddocks. Also, to show how accurate the image georeferencing was, we have attached the reported accuracy and error by Pix4D for the GCP points. As you can see, the error on Z coordinate is very low.

  1. Discussion: I think that this research still requires some additional discussion. Please compare your results (incl. research design) with those from other studies. It might be interesting to reflect on the implementation of other sensors (e.g. LiDAR), and on the fusion of UAV and satellite imagery (and the other suggestions you mentioned in the last part of the Conclusion).

Thanks a lot for your comment. The further discussion has been added to section 4.3.

Reviewer 2 Report

Comments and Suggestions for Authors

Dear authors,

I want to congratulate you for the work done for this research, but I want to bring some improvements to this work.

First of all, the Introduction section should perhaps be slightly restructured, to be more concise. At the same time, the purpose and objectives of the work of this publication should emerge and what is the novelty of this publication. It is true that the purpose is specified in the last part of the introduction, but this part should be developed along with the novelty of this study, there are quite a lot of studies in this field.

Other questions/improvements:

2.1 Why was the study carried out only in one year (2022), we didn't need maybe 2-3 experimental years?

Line 128 – perhaps the type of meadows existing on the pasture should be specified

2.2 - grazing was done only with cattle, and what is the load of LU/ha?

Line 144 – how were the 3 specific dates decided? Is it the optimal time for plants?

Table 1 – line 166 – how was Flight Altitude (60) chosen?? Maybe it was interesting to include the time of flight, that is, the flights were made in broad daylight, full light, morning, or evening, etc.

Line 355 – what was the grass recovery time?

Line 524 – why was the unit of measure g/m chosen?

3.1.2. Image Gridding – why 1m/1m variants were chosen?

Author Response

Reviewer 2:

Dear Reviewer,

We are grateful for your valuable feedback and thoughtful comments on our manuscript titled "Pasture Biomass Estimation Using Ultra High-Resolution RGB UAVs Images and Deep Learning." Your insights are greatly appreciated, and we have addressed your comments and suggestions as follows:

  1. First of all, the Introduction section should perhaps be slightly restructured, to be more concise. At the same time, the purpose and objectives of the work of this publication should emerge and what is the novelty of this publication. It is true that the purpose is specified in the last part of the introduction, but this part should be developed along with the novelty of this study, there are quite a lot of studies in this field.

Thanks a lot for your comment. We restructured the introduction based on your comment.

  1. Why was the study carried out only in one year (2022), we didn't need maybe 2-3 experimental years?

In response to the reviewer's question regarding the study's duration, we acknowledge the common practice of conducting agricultural studies over 2-3 years to account for variations. However, the rationale for the one-year study duration in our case was centered on the specific objectives of our research.

Our study focused on the integration of machine learning and drone imagery for biomass estimation. The primary emphasis was on the development of a robust model based on a substantial dataset. To achieve this, we concentrated on collecting a large and comprehensive dataset in a single year. This allowed us to ensure that our model was built on a sufficiently diverse and representative set of data, which is essential for its robustness and reliability.

While the study was conducted in one year, we fully intend to validate and test the model on new datasets in the future to assess its performance and generalizability across different years and conditions. This approach aligns with the standard practice in agricultural research, and we plan to incorporate multi-year data in future publications to further demonstrate the model's effectiveness and adaptability.

  1. Line 128 – perhaps the type of meadows existing on the pasture should be specified

Thanks a lot for your comment. These pastures are made up of predominantly Tall Fescue cool season grass. We added it to the manuscript.

  1. grazing was done only with cattle, and what is the load of LU/ha?

Thanks a lot for your comment. Paddocks were 0.81 ha each and were rotationally grazed by 8 cows, with cows moved to the next paddock when grass and/or hay had been grazed.

  1. Line 144 – how were the 3 specific dates decided? Is it the optimal time for plants?

Thanks a lot for your comment. These three dates corresponded to immediately after the cattle finished winter hay feeding (Mid-April), followed by sampling in Mid-May when the grass had partially regrown, and then lastly in early June when the grass should have fully regrown.

  1. Table 1 – line 166 – how was Flight Altitude (60) chosen?? Maybe it was interesting to include the time of flight, that is, the flights were made in broad daylight, full light, morning, or evening, etc.

Regarding the choice of flight altitude (60 ft) in our study, we appreciate the reviewer's insightful comment. The decision to fly at this altitude was indeed influenced by several factors, including drone capabilities and environmental conditions. We used a small Mavic Pro drone for our data collection, which has certain limitations, particularly in windy conditions. Attempting to fly at a higher altitude in a windy area could lead to stability and safety issues, as indicated by the drone's error messages. Therefore, to ensure the safety and stability of the data collection process, we had to make the trade-off of flying at a lower altitude.

About the timing of our drone flights, we conducted our flights specifically during local noon. This timing was chosen to ensure optimal lighting conditions for image acquisition. Our flights took place in both full sunny sky and overcast conditions.

We added this information to this section.

  1. Line 355 – what was the grass recovery time?

Thanks a lot for your comment. We are not sure how to respond. That's what we're measuring, as explained in comment 3 above for grass regrowth

  1. Line 524 – why was the unit of measure g/m chosen?

Thanks a lot for your comment. The choice of the unit of measure "g/m" was made to ensure consistency with existing literature and standards in the field. This unit, representing grams per meter, is commonly used for expressing quantities related to biomass, especially in studies involving vegetation or ecological assessments. It provides a clear and standardized way to convey the amount of biomass per unit length, facilitating comparability with other research findings and enabling broader contextualization within the scientific community. both lb and g are valid units for expressing biomass, we made a deliberate choice to adopt the unit in the International System of Units (SI), which is grams per meter (g/m).

  1. Image Gridding – why 1m/1m variants were chosen?

We appreciate the reviewer's inquiry regarding the rationale behind the chosen image gridding resolution at 1m² and the ground sampling quadrat dimensions of 0.25 m² (as mentioned in Line 142). In the existing body of literature, numerous investigations have employed a ground sampling size of 0.25m² as a widely adopted measure for forage cutting. Nevertheless, our aim was to identify a standardized and optimal size that could effectively streamline the computational processes involved in map generation while preserving all pertinent details. The decision to use a 1m² image gridding resolution was driven by the need to strike a balance between capturing detailed information about the study area and maintaining computational efficiency. A finer resolution may result in an excessively large dataset, posing challenges in terms of computational resources and processing time. On the other hand, a coarser resolution might lead to a loss of critical spatial information. The 1m² resolution was deemed optimal for our study as it allowed us to sufficiently capture relevant spatial patterns and features while managing computational demands effectively.

Reviewer 3 Report

Comments and Suggestions for Authors

The manuscript is well-structured, and the logic is easy to follow. The coefficient of determination is relatively high for the three methods compared, favoring the use of ANN. Nonetheless, high values for R2 and low values for MAE and RMSE do not seem to coincide with the spread of points shown in the X=Y plot. There might be several reasons associated with this mismatch. It is impossible to suggest potential causes without the original data. To improve the presentation of the case, let me recommend:

1. Include an analysis of the residuals (e.g., plot residuals versus X and Y, plot residuals versus time, normal order statistics)

2. Repeated measure analysis is warranted since the data points were collected over three dates.

3. Show the metrics to ascertain how reliable the estimation of PH is. With the apparent variability in estimating PH, it seems that the high R2 is forced.

4. How accurate is the estimation of the spatial variability? This issue needs to be discussed further. It is of critical importance in applying the method for decision-making.

Please ensure the numbers for observed biomass values in Figure 6 are shown correctly.

Author Response

Dear Reviewer,

We are grateful for your valuable feedback and thoughtful comments on our manuscript titled "Pasture Biomass Estimation Using Ultra High-Resolution RGB UAVs Images and Deep Learning." Your insights are greatly appreciated, and we have addressed your comments and suggestions as follows:

  1. Include an analysis of the residuals (e.g., plot residuals versus X and Y, plot residuals versus time, normal order statistics)

Thanks a lot for your comment. We added those plots to the manuscript.

  1. Repeated measure analysis is warranted since the data points were collected over three dates.

Thanks a lot for your comment. We edited those plots.

  1. Show the metrics to ascertain how reliable the estimation of PH is. With the apparent variability in estimating PH, it seems that the high R2 is forced.

Thanks a lot for your comment. We edited those plots.

  1. How accurate is the estimation of the spatial variability? This issue needs to be discussed further. It is of critical importance in applying the method for decision-making.

Thanks a lot for your comment. We added it to the discussion part.

Round 2

Reviewer 1 Report

Comments and Suggestions for Authors

The authors addressed all my comments and provided the necessary additional clarification.

Author Response

Thanks a lot for your time and valuable comments